# The Effect of Soil Sampling Density and Spatial Autocorrelation on Interpolation Accuracy of Chemical Soil Properties in Arable Cropland

Dorijan Radočaj [1,*], Irena Jug [1], Vesna Vukadinović [1], Mladen Jurišić [1] and Mateo Gašparović [2]

1 Faculty of Agrobiotechnical Sciences Osijek, Josip Juraj Strossmayer University of Osijek, Vladimira Preloga 1, 31000 Osijek, Croatia; irena.jug@fazos.hr (I.J.); vesna.vukadinovic@fazos.hr (V.V.); mjurisic@fazos.hr (M.J.)
2 Faculty of Geodesy, University of Zagreb, Kačićeva 26, 10000 Zagreb, Croatia; mgasparovic@geof.unizg.hr
* Correspondence: dradocaj@fazos.hr; Tel.: +385-31-554-965

**Abstract:** Knowledge of the relationship between soil sampling density and spatial autocorrelation with interpolation accuracy allows more time- and cost-efficient spatial analysis. Previous studies produced contradictory observations regarding this relationship, and this study aims to determine and explore under which conditions the interpolation accuracy of chemical soil properties is affected. The study area covered 823.4 ha of agricultural land with 160 soil samples containing phosphorus pentoxide ($P_2O_5$) and potassium oxide ($K_2O$) values. The original set was split into eight subsets using a geographically stratified random split method, interpolated using the ordinary kriging (OK) and inverse distance weighted (IDW) methods. OK and IDW achieved similar interpolation accuracy regardless of the soil chemical property and sampling density, contrary to the majority of previous studies which observed the superiority of kriging as a deterministic interpolation method. The primary dependence of interpolation accuracy to soil sampling density was observed, having $R^2$ in the range of 56.5–83.4% for the interpolation accuracy assessment. While this study enables farmers to perform efficient soil sampling according to the desired level of detail, it could also prove useful to professions dependent on field sampling, such as biology, geology, and mining.

**Keywords:** spatial autocorrelation; ordinary kriging; inverse distance weighted; prediction; mapping

## 1. Introduction

Spatial interpolation of soil chemical and physical properties is necessary to model its continuous distribution from discrete geo-referenced soil samples, which in this form do not exhibit a representative state of the agricultural land [1]. Monitoring the spatio-temporal dynamics of soil parameters is necessary for sustainable agricultural land management due to their heterogeneity affected by edaphic processes and agricultural production systems, which are difficult to record and model [2]. Detection of input parameters for spatial interpolation, such as sampling density and method as well as terrain heterogeneity, enables more economical and efficient soil sampling by adjusting the sampling plan to accommodate these factors [3]. The influence of these segments with varying intensity affects the heterogeneity of agricultural land and can be divided into micro- and macro-level [4]. The micro-level includes one or several neighboring agricultural parcels, while the macro-level covers administrative units ranging from municipalities to the state level. One of the more important applications of spatial interpolation of soil parameters at the micro-level is mapping agriculture in precision [5], which affects farmers' financial gain and environmental protection due to the reduced application of mineral fertilizers and pesticides [6]. Conducted studies at the macro-level are aimed at better decision-making related to spatial planning and management of agricultural land. Determining the level of impact of soil sampling density is important for both levels of research, given the high cost and time inefficiency of conventional field sampling and laboratory soil analysis [7].

According to the Web of Science Core Collection (WoSCC) database, the number of scientific papers published from 2010 to 2020 indicates a steady growth in soil properties prediction studies (Figure 1). The query consisted of the combination of the term "soil", combined with the terms "interpolation" or "prediction", and terms specific for macro- and micro-level research. Among the most commonly used methods, geostatistical and deterministic interpolation methods have been commonly applied in these studies over the past decade. The application of machine learning to predict soil parameters has grown rapidly since 2017 but requires a substantial number of covariates, being time inefficient on smaller study areas.

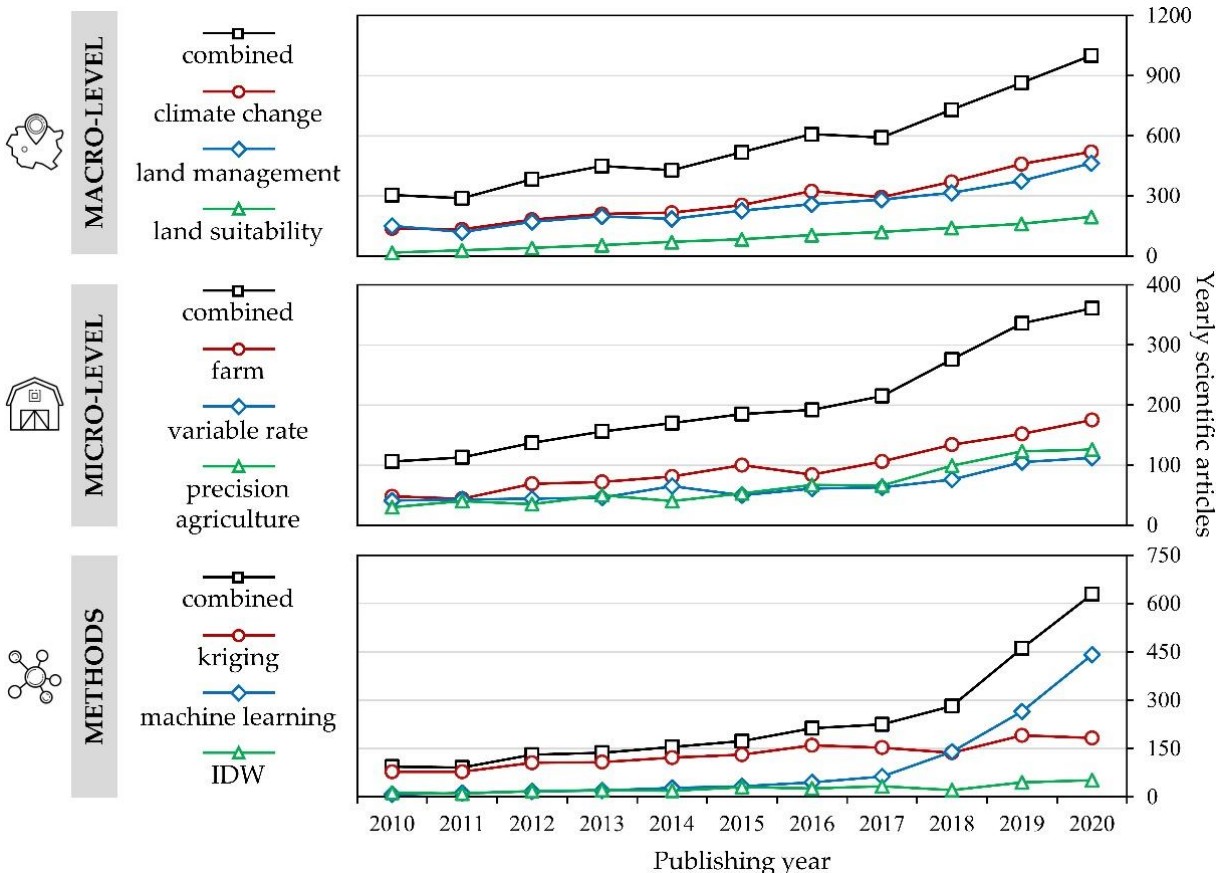

**Figure 1.** The number of scientific articles indexed in WoSCC from 2010–2020, based on the combination of terms "soil", "interpolation", or "prediction", with specific terms listed.

Numerous studies have focused on selecting optimal soil sampling methods, the hypothesis being that a larger number of soil samples allows for proportionally higher interpolation accuracy being fully or mostly accepted (Table 1). One of the main reasons for the difference in the influence of the number of soil samples on the interpolation accuracy was the average area per sample, though it was not possible to clearly define the degree of its influence on the interpolation accuracy. Studies in which a low dependence of soil sampling density on the accuracy of spatial interpolation was found have shown that a more important parameter is the level of spatial autocorrelations between samples. Land cover classes and topographic properties of sampled areas also impacted interpolation accuracy [8]. Liu et al. [9] noted a low degree of dependence of interpolation accuracy on the number of collected samples under conditions of moderate spatial autocorrelation at the macro-level. Instead, high dependence of interpolation accuracy on land cover class, precipitation, and air temperature was noted, parameters that are variable only at the macro-level. Long et al. [7] also noted a higher dependence of interpolation accuracy on sampling density for a lowland terrain in a large study area. Similar observations

were obtained at the micro-level of research, while the importance of land cover was much more impactful. Its inclusion in the study allowed the increase of interpolation accuracy and reduced the number of required samples by approximately 6% compared to the conventional approach [10]. At the micro-level, increasing the spatial autocorrelations according to the land cover class is appropriate, indicating that sampling density is not necessarily the primary factor of spatial interpolation accuracy. Previous research at the micro-level more strongly indicates the primary importance of spatial autocorrelation in determining the accuracy of spatial interpolation. Li [3] analyzed soil sampling densities in the range of the smallest distance between adjacent samples of 25 m to 500 m and found that the sparse sampling distance of 250 m resulted in the highest interpolation accuracy. At the same time, the distribution and the coefficients of variation of the applied data sets did not differ significantly from each other. Based on similar research, Kravchenko [11] concluded that the level of spatial autocorrelation, regardless of the coefficient of variation, is the primary factor in the accuracy of spatial interpolation. Rodrigues et al. [12] noted the dependence of interpolation accuracy on sampling density only in cases below 50% of the collected samples, with only a very small sample count producing a negative effect on the interpolation accuracy. Quantification of spatial autocorrelation in spatial interpolation studies is most often performed using the Moran's I index, which determines the complex relationships of soil sample values in their environment [13].

**Table 1.** Literature review of the effect soil sampling density in agricultural land has on spatial interpolation accuracy.

| Reference | Study Area | Total Sample Count | Average Area per Sample (ha) | Correlation of Interpolation Accuracy and Sampling Density |
|---|---|---|---|---|
| Rodrigues et al. [12] | 72 ha | 4306 | 0.02 | low |
| Kravchenko [11] | 20 ha | 529 | 0.04 | low |
| Zhang et al. [14] | 72 km$^2$ | 2755 | 2.61 | moderate |
| Zhang et al. [10] | 40 km$^2$ | 997 | 4.01 | high |
| Long et al. [7] | 10,636 km$^2$ | 188,247 | 5.65 | high |
| Zhang et al. [15] | 40 km$^2$ | 214 | 18.7 | high |
| Shen et al. [1] | 173 km$^2$ | 700 | 24.7 | high |
| Li [3] | 400 km$^2$ | 335 | 119 | low |
| Sun et al. [16] | 683 km$^2$ | 394 | 173 | high |
| Zhao et al. [17] | 1450 km$^2$ | 745 | 195 | moderate |
| Ye et al. [18] | 16,400 km$^2$ | 1458 | 1125 | high |
| Liu et al. [9] | 620,000 km$^2$ | 382 | 162,304 | low |

Accounting for the inconsistency of previous research on the dependence of soil sampling density and spatial interpolation accuracy, the research hypothesis is that interpolation accuracy primarily depends on the spatial autocorrelation of input values. The primary goal of the research was to determine the validity of the hypothesis by applying different densities of soil samples in the same research area and observing their impact on interpolation accuracy. Secondary objectives were to determine the applicability of the Moran's I index as a value for quantifying spatial autocorrelation, to predict interpolation accuracy, and to determine which of the interpolation methods used is most appropriate for evaluating soil samples.

## 2. Materials and Methods

### 2.1. Study Area

The study area covers 823.4 ha in Osijek-Baranja County in eastern Croatia (Figure 2). It is mainly a lowland area, with an average altitude of 88 m. The dominant land cover is non-irrigated, arable agricultural land (code 211), according to the CORINE 2018 land cover classification. This area is traditionally a hotspot for agricultural production in the Republic of Croatia, with maize, wheat, and sunflower as the major cultivated crops [19]. Based on previous research in eastern Croatia, it was noted that the existing natural resources are

not used optimally according to agricultural land management plans [20]. Per FAO-85 soil classification of the European Soil Database v2.0 database, pseudogley Luvisol (Lo) covers the entire study area. The climate is moderately warm and rainy, representing class Cfwbx of the Köppen classification [21].

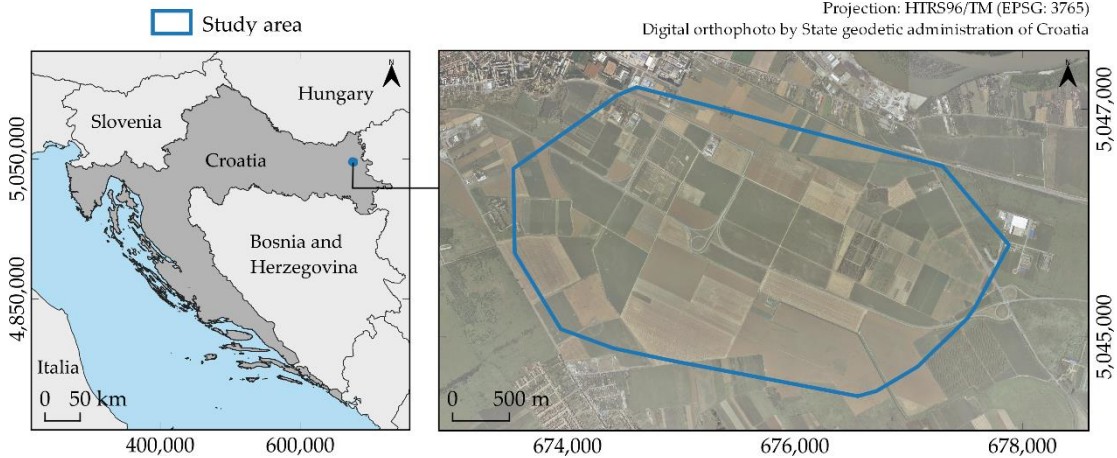

**Figure 2.** Study area and its location within the Republic of Croatia.

### 2.2. Soil Sampling Data

A total of 160 soil samples were used in the study, based on 20 soil cores at a depth of 0–30 cm within a field. The samples were provided by Osijek-Baranja County, with an average sampling density of 5.1 ha per sample. Sampling was performed using the random sampling method. Laboratory analysis using the Al method [22] was used to determine the soil phosphorus pentoxide ($P_2O_5$) and potassium oxide ($K_2O$) contents, expressed in mg 100 $g^{-1}$. The reliable spatial representation of $P_2O_5$ and $K_2O$, as the macro-nutrients in the soil, are traditionally important in sustainable agriculture, which is further intensified by the emergence of the concept of precise fertilization [6]. Monitoring the dynamics of $P_2O_5$ and $K_2O$ at the micro-level enables quality management of agricultural land and environmental protection, enabling the optimal application of mineral fertilizer [23,24]. At the macro-level, it enables monitoring and remediation of soil degradation [25].

The original soil sample set was split using a geographically stratified random splitting procedure to create soil sample subsets. The stratification was performed by horizontally and vertically splitting the study area into four equal zones. Samples in each of these zones were then randomly split in eight variations according to the percentage of the original soil sample set: 100%, 87.5%, 75%, 62.5%, 50%, 37.5%, 25%, and 12.5%. After the splits within each zone, corresponding subsets containing the same percentages were merged into a study area subset covering the entire study area, forming eight subsets in total (Figure 3). This approach achieved a balanced spatial distribution of soil samples over an entire study area, which more accurately simulates the soil sampling procedure in the field than the conventional random split procedure. A similar variation of the applied method for creating soil sample subsets was successfully performed in studies by Kravchenko [11] and Rodrigues et al. [12].

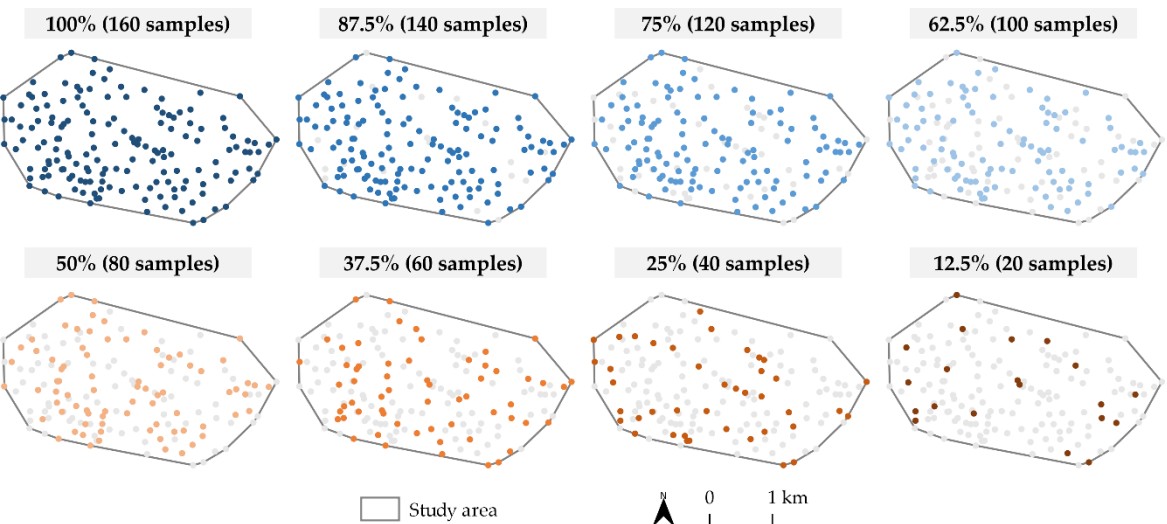

**Figure 3.** The display of soil samples contained in eight soil subsets.

The basic properties of the input soil sample subsets were evaluated using descriptive statistics consisting of arithmetic mean, coefficient of variation (CV), and minimum and maximum values. The spatial autocorrelation of the input soil sample subsets was evaluated using a correlogram and a univariate global Moran's I index [26]. Positive values of Moran's I indicate a proportionally positive spatial autocorrelation, while negative values indicate a lack of spatial autocorrelation [27]. Moran's I was calculated according to Formula (1):

$$\text{Moran's I} = \frac{n}{\sum_{i=1}^{n}\sum_{j=1}^{n} w_{MI}} \cdot \frac{\sum_{i=1}^{n}\sum_{j=1}^{n} w_{MI}(y_i - \overline{y})(y_j - \overline{y})}{\sum_{i=1}^{n}(y_i - \overline{y})^2}, \tag{1}$$

where $n$ represents a quantity of soil samples per subset, $w_{MI}$ represents spatial weight which indicates a spatial relationship of two neighboring samples, $y_i$ represents sampled $P_2O_5$ and $K_2O$ values, while $\overline{y}$ represents the arithmetic mean of all input values per subset. Distance weights for Moran's I were determined using the K-nearest neighbors method, based on the four neighbors of each soil sample. The optimal spatial resolution of the interpolation results relates to the number of samples per subset and was determined for each of the eight subsets using the Inspection Density method according to Formula (2) by Hengl [28]:

$$r = 0.0791 \cdot \sqrt{\frac{A}{n}}, \tag{2}$$

where $r$ is the spatial resolution and $A$ is the study area. The open-source GIS software SAGA GIS v.7.3.0 was used for spatial interpolation and assessment of interpolation accuracy, GeoDa v1.18 was used to assess spatial autocorrelation, and QGIS v3.8.3 was used for the visualization of interpolation results. All spatial calculations were performed in the coordinate reference system HTRS96/TM.

### 2.3. Spatial Interpolation Methods and Interpolation Parameters

Ordinary kriging (OK) and the inverse distance weighted (IDW) were selected as spatial interpolation methods to analyze the effect of spatial autocorrelation and sampling density on interpolation accuracy. Previous research has found that these methods achieve optimal interpolation accuracy under different conditions of normal distribution and stationarity of input data, being complementary in various cases [29]. Although OK is in most cases a superior method of spatial interpolation in terms of prediction accuracy [1], IDW supports the very few soil samples for the interpolation, which can disable variograms fitting for OK [20]. OK has been declared the best linear unbiased spatial interpolation

method in the case of normal distribution of input data, being also the most commonly used geostatistical interpolation method for various soil parameters [30]. Though it is one of the most efficient and flexible interpolation methods, it requires the normal distribution and stationarity of input data [31]. OK uses a variogram to model spatial autocorrelation under the assumption that the observed soil parameter at a particular location is similar to closer samples in proportion to their mutual distance [9]. Formula (3) was used to model the variogram:

$$\gamma(d) = \frac{1}{n(d)} \sum_{i=1}^{n(d)} [z(x_i) - z(x_i+d)]^2, \tag{3}$$

where $\gamma(d)$ represents a variance at the distance $h$, $n(d)$ represents a number of lags at distance $d$, $z(x_i)$ and $z(x_i + d)$ represent soil sampling values at the locations $x_i$ i $x_i + d$. Prediction of the $P_2O_5$ and $K_2O$ values at the unknown locations was performed based on the variogram according to the Formula (4):

$$z_{OK} = \sum_{i=1}^{n} \lambda_i \cdot z(x_i), \tag{4}$$

where $z_{OK}$ represents interpolated value using the OK and $\lambda_i$ represents weight determined using the variogram. The presence of a normal distribution of $P_2O_5$ and $K_2O$ values in the soil was examined using the Shapiro–Wilk test, using R software v4.0.3. In the absence of normal distribution cases, the logarithmic transformation of the input values was performed as a preprocess for OK interpolation. Spatial autocorrelation in variogram modeling was evaluated at a distance of 1200 m from at least 15 known points, including 12 lags, each of which covered a distance of 100 m. The tested mathematical models for variogram fitting were linear, square root, power, Gaussian, and spherical models, explained in detail in [32]. The selection of the optimal mathematical model was performed according to the highest level of fitting the mathematical model to the variogram, expressed by the coefficient of determination ($R^2_v$). For each set of input data, the basic parameters of the selected mathematical model were examined, including the nugget (n), sill (s), and range (r).

IDW belongs to the most commonly used deterministic spatial interpolation methods, characterized by ease of application due to the small number of interpolation parameters [11]. Prediction of $P_2O_5$ and $K_2O$ values at an unknown location was performed using weighted inverse distances and sampled soil values according to Formula (5):

$$z_{IDW} = \frac{\sum_{i=1}^{n} z(x_i) \cdot d^{-w_{IDW}}}{\sum_{i=1}^{n} d^{-w_{IDW}}}, \tag{5}$$

where $w_{IDW}$ represents the weight calculated using the inverse distance from the soil samples to an unknown location. The inverse distance exponent for interpolation was 3, the most suitable value determined based on the iterative interpolation procedure for all input subsets. The maximum distance for the prediction corresponds to the distance selected for OK interpolation and equals 1200 m, with at least 15 neighboring samples used for the prediction.

### 2.4. Interpolation Accuracy Assessment and Relationship with Sampling Density and Spatial Autocorrelation

The accuracy assessment of the OK and IDW interpolation results for the 100% subset was performed by cross-validating the input values using the leave-one-out technique. The partial subsets were evaluated according to the soil samples excluded from the original soil sample set, implementing a split-sample validation to allow more comprehensive assessment accuracy. The coefficient of determination ($R^2$), root mean square error (RMSE), and normalized root mean square error (NRMSE) were metrics used for the accuracy assessment. $R^2$ and RMSE allow a comprehensive analysis of the interpolation accuracy

due to their complementarity [33]. NRMSE resulted in a relative interpolation error value and allowed a parallel interpolation accuracy assessment of both soil parameters with different value intervals [7]. These values were calculated according to Formulas (6)–(8):

$$R^2 = 1 - \frac{\sum_1^n (y_i - \widetilde{y}_i)^2}{\sum_1^n (y_i - \overline{y})^2} \tag{6}$$

$$RMSE = \sqrt{\frac{\sum_1^n (y_i - \widetilde{y}_i)^2}{n}}, \tag{7}$$

$$NRMSE = \frac{RMSE}{\overline{y}}, \tag{8}$$

where $\widetilde{y}_i$ represents interpolated $P_2O_5$ and $K_2O$ values. Sampling density and Moran's I values for both interpolation methods were compared with the interpolation accuracy results represented by $R^2$ and RMSE. The relationship of the spatial autocorrelation and soil sampling density to interpolation accuracy was modeled using linear regression separately for $P_2O_5$ and $K_2O$. The strength of their dependence was quantified proportionally by the coefficient of determination.

## 3. Results

A high range of $P_2O_5$ values in the soil in the study area was observed, while $K_2O$ values showed low variability (Table 2). For both soil chemical properties, CV values retained a close value range, which began to increase for sampling densities of less than 37.5% of the original soil samples. The $p$ values of the Shapiro–Wilk test resulted in values below 0.05 for five subsets with the highest percentages of soil samples for $P_2O_5$, as well as the top seven subsets for $K_2O$. The null hypothesis of normal data distribution for these subsets was rejected, and logarithmic transformation as a preprocess for OK was performed. The spatial resolution determined by the Inspection Density method showed a relatively low difference of between 100% and 50% of soil samples, dropping off more intensively for sparser subsets.

**Table 2.** Descriptive statistics and Shapiro–Wilk test results of soil subsets.

| Soil Property | Percentage of Soil Samples | Mean | CV | Min | Max | Shapiro–Wilk | | Target Spatial Resolution (m) |
|---|---|---|---|---|---|---|---|---|
| | | | | | | W | p | |
| $P_2O_5$ | 100% | 21.59 | 0.32 | 8.3 | 36.5 | 0.971 | 0.002 | 18 |
| | 87.5% | 21.59 | 0.31 | 8.3 | 36.5 | 0.973 | 0.007 | 19 |
| | 75% | 21.44 | 0.32 | 10.3 | 36.5 | 0.968 | 0.006 | 21 |
| | 62.5% | 21.55 | 0.32 | 10.5 | 36.5 | 0.967 | 0.012 | 23 |
| | 50% | 20.75 | 0.31 | 10.5 | 35.0 | 0.963 | 0.022 | 25 |
| | 37.5% | 21.65 | 0.33 | 8.3 | 36.5 | 0.972 | 0.180 | 29 |
| | 25% | 22.01 | 0.33 | 8.3 | 36.5 | 0.965 | 0.236 | 36 |
| | 12.5% | 21.55 | 0.39 | 10.5 | 36.5 | 0.936 | 0.198 | 51 |
| $K_2O$ | 100% | 24.43 | 0.15 | 16.7 | 34.4 | 0.944 | >0.001 | 18 |
| | 87.5% | 24.49 | 0.15 | 16.7 | 34.2 | 0.942 | >0.001 | 19 |
| | 75% | 24.36 | 0.15 | 16.7 | 34.4 | 0.952 | >0.001 | 21 |
| | 62.5% | 24.28 | 0.15 | 16.7 | 33.6 | 0.945 | >0.001 | 23 |
| | 50% | 24.82 | 0.15 | 19.5 | 34.4 | 0.938 | 0.001 | 25 |
| | 37.5% | 24.67 | 0.16 | 17.2 | 34.4 | 0.937 | 0.004 | 29 |
| | 25% | 24.62 | 0.18 | 17.2 | 34.2 | 0.923 | 0.008 | 36 |
| | 12.5% | 24.02 | 0.18 | 17.2 | 34.4 | 0.935 | 0.192 | 51 |

Moran's I values of all subsets revealed a positive, moderately-high spatial autocorrelation (Figure 4). A stable Moran's I was retained while lowering soil sample percentages by up to 25% of the original datasets, the values dropping noticeably for the 12.5% subset. Both soil chemical properties resulted in values of over 0.500 for these subsets, while $K_2O$ resulted in a slightly higher spatial autocorrelation than $P_2O_5$ values. The distance of the spatial autocorrelation per subset was analyzed by correlograms, showing stable value and a noticeable increase for the 25% and 12.5% subsets (Appendix A, Figure A1).

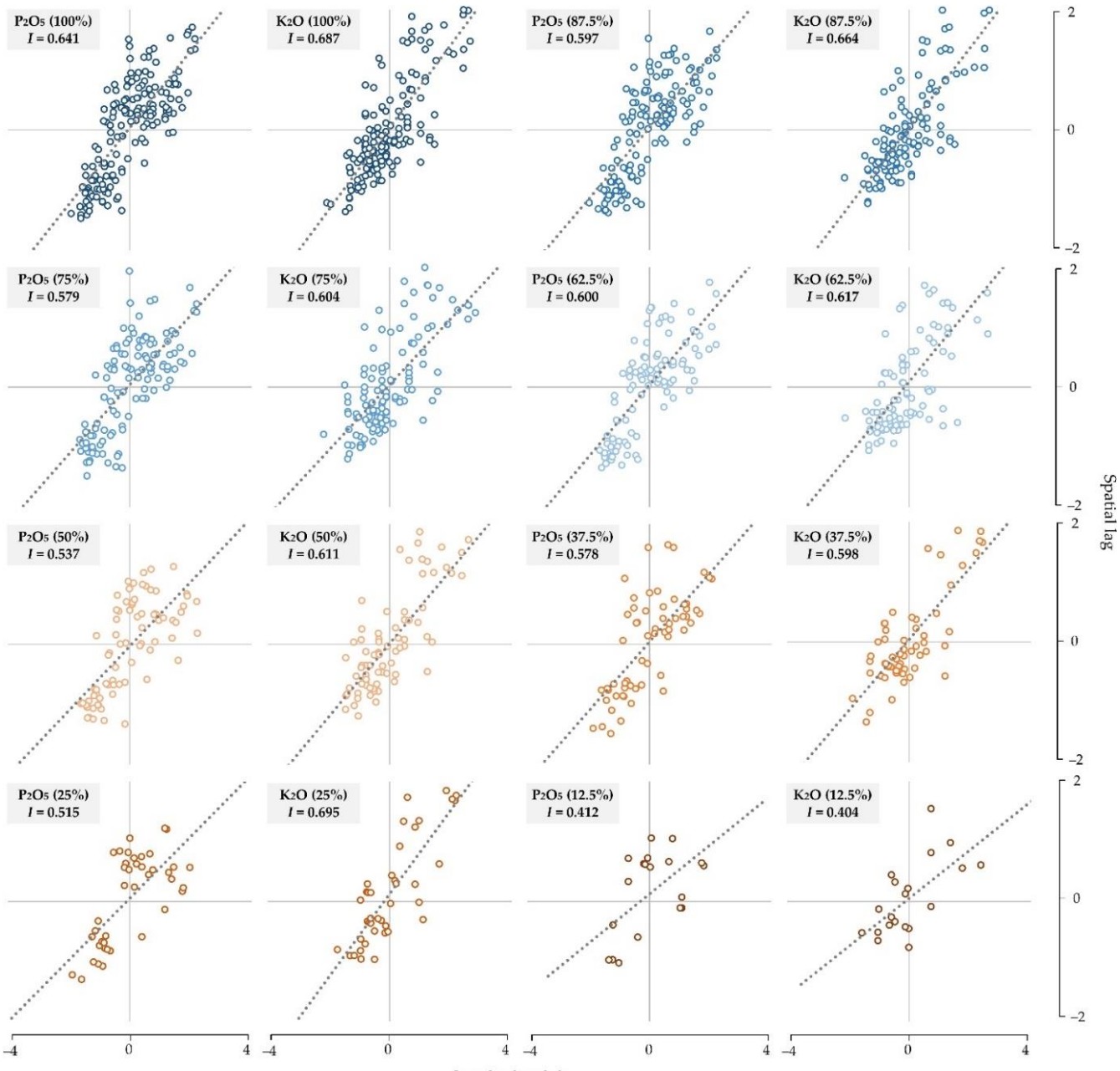

**Figure 4.** Moran's I values of the eight soil subsets for $P_2O_5$ and $K_2O$.

The optimal OK interpolation parameters per subset are shown in Appendix A, Table A1. The mathematical power model was determined as the best fit for all interpolation variants. Fitting autocorrelation ranges corresponded to those determined by the correlograms, increasing for 25% and 12.5% soil subsets. The highest $R^2_v$ values were observed for denser soil subsets, especially for 62.5% and higher percentages of the original

soil samples. Both OK and IDW achieved high spatial interpolation accuracy for subsets containing 37.5% or more of the original samples (Table 3). The $P_2O_5$ was more accurately predicted by IDW in six cases, per RMSE and NRMSE values. IDW achieved maximum interpolation accuracy for the 87.5% subset, followed by the full soil subset using the OK. Considering the $R^2$ values for $P_2O_5$, the top two subsets produced a large gap in the results for 75%, 62.5%, 50%, and 37.5% values, while the bottom two subsets further showed a considerable decline in interpolation accuracy. OK was a more accurate interpolation method in seven of eight cases for the $K_2O$ samples, with higher spatial autocorrelation than the $P_2O_5$ values. Similar to the previous case, the most accurate interpolation variant was achieved using the 100% soil samples, with accuracy significantly dropping for the two sparsest subsets.

**Table 3.** Interpolation accuracy results for soil subsets using OK and IDW interpolation methods.

| Soil Property | Percentage of Soil Samples | OK | | | IDW | | |
|---|---|---|---|---|---|---|---|
| | | $R^2$ | RMSE | NRMSE | $R^2$ | RMSE | NRMSE |
| $P_2O_5$ | 100% | 0.743 | **4.157** | **0.193** | 0.713 | 4.249 | 0.197 |
| | 87.5% | 0.729 | 4.272 | 0.198 | **0.751** | 4.211 | 0.195 |
| | 75% | 0.628 | 4.468 | 0.208 | 0.653 | 4.308 | 0.201 |
| | 62.5% | 0.630 | 4.696 | 0.218 | 0.623 | 4.466 | 0.207 |
| | 50% | 0.618 | 4.702 | 0.227 | 0.614 | 4.526 | 0.218 |
| | 37.5% | 0.581 | 4.394 | 0.203 | 0.687 | 4.323 | 0.202 |
| | 25% | 0.445 | 5.135 | 0.233 | 0.449 | 5.182 | 0.235 |
| | 12.5% | 0.487 | 5.190 | 0.241 | 0.492 | 5.044 | 0.234 |
| $K_2O$ | 100% | **0.794** | **2.080** | **0.085** | 0.759 | 2.172 | 0.089 |
| | 87.5% | 0.774 | 2.127 | 0.087 | 0.704 | 2.473 | 0.101 |
| | 75% | 0.760 | 2.127 | 0.087 | 0.716 | 2.325 | 0.095 |
| | 62.5% | 0.727 | 2.324 | 0.096 | 0.668 | 2.438 | 0.100 |
| | 50% | 0.688 | 2.884 | 0.116 | 0.634 | 2.457 | 0.099 |
| | 37.5% | 0.637 | 2.275 | 0.173 | 0.629 | 2.327 | 0.094 |
| | 25% | 0.455 | 2.678 | 0.109 | 0.469 | 2.702 | 0.110 |
| | 12.5% | 0.518 | 2.751 | 0.115 | 0.508 | 2.703 | 0.113 |

The highest interpolation accuracy values per soil property are in bold type.

The OK and IDW interpolation results using all eight subsets with $P_2O_5$ and $K_2O$ values are shown in Figure 5. All interpolation results predicted the highest state of soil $P_2O_5$ in the northern and western parts of the study area, with variable levels of local heterogeneity. The highest concentration of $K_2O$ was observed in the central and northern part of the study area, declining along the borders of the study area to low intensity. The wide value range of both $P_2O_5$ and $K_2O$ was retained from the input values of soil samples in the interpolation results, primarily in the case of IDW. IDW retained a nearly constant level of local soil $P_2O_5$ heterogeneity, dropping off only in a sample density below 50%. For OK, the level of heterogeneity of the interpolation results is more uniform, due to the same applied mathematical model and similar *n* and *s* values.

Both $R^2$ and RMSE representing the interpolation accuracy of the combined OK and IDW results indicated a strong correlation to the sampling density (Figure 6). Sampling density resulted as a prime indicator of the interpolation accuracy, having a superior correlation with $R^2$ and RMSE for both soil chemical properties. The spatial autocorrelation represented by Moran's I showed a lower impact on the interpolation accuracy, with a slightly higher impact on lower spatial autocorrelation $P_2O_5$ values.

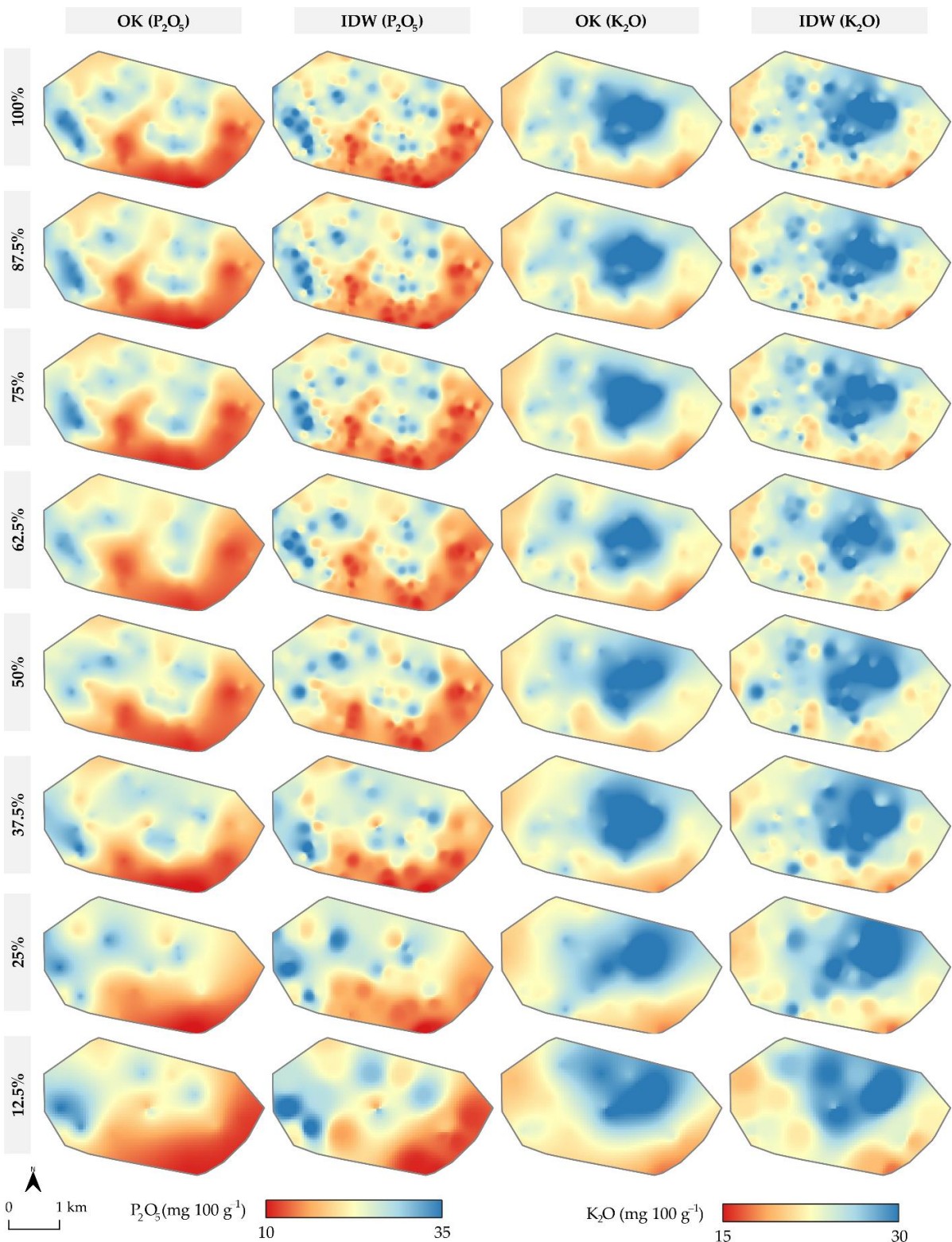

**Figure 5.** Interpolation results for the combination of eight soil subsets with $P_2O_5$ and $K_2O$ values.

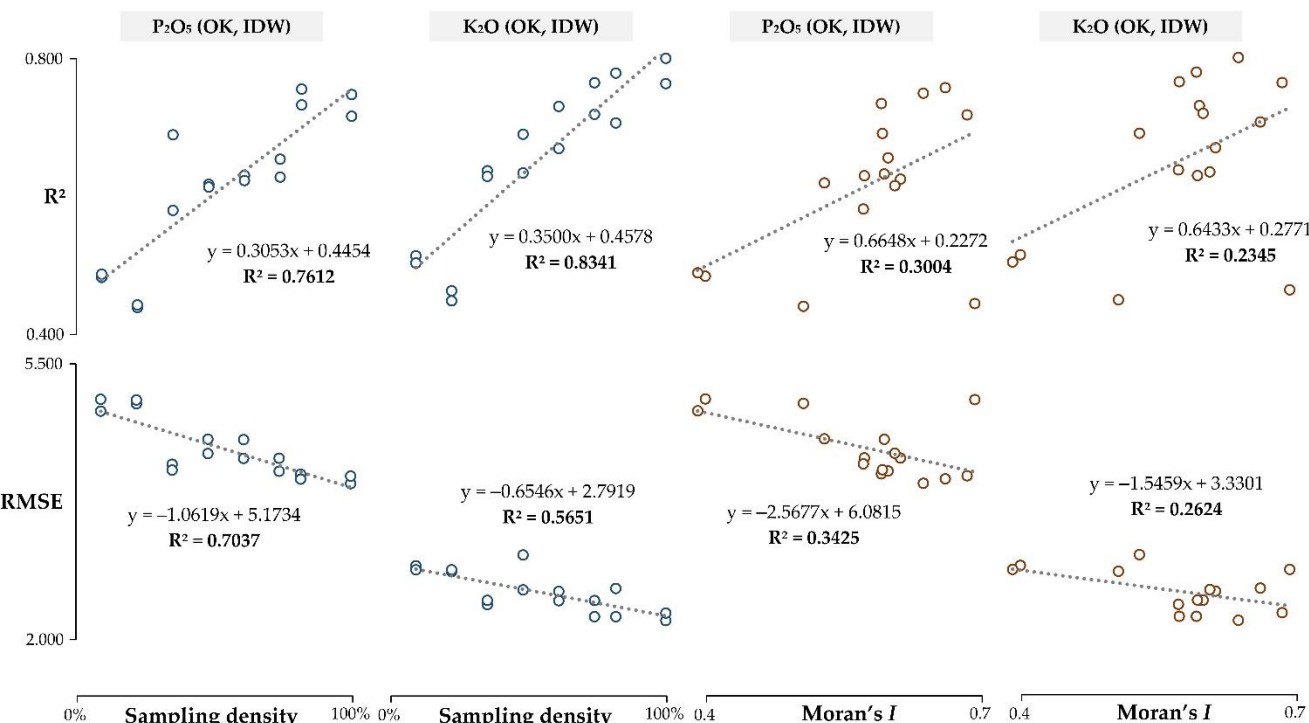

**Figure 6.** The relationship of the spatial autocorrelation and sampling density with interpolation accuracy.

## 4. Discussion

The area per sample of 5.1 ha for the full soil sample set in this study indicates its primary importance towards understanding the relationship between sampling density and spatial autocorrelation with interpolation accuracy on a micro-level. The main observation of this study is the predominant impact of the sampling density on interpolation accuracy, achieving high $R^2$ for both $P_2O_5$ and $K_2O$ values. This aligns with the observations of previous studies with a similar average area per sample, where interpolation accuracy achieved a high correlation with sampling density [7,10]. The exception to this statement related to previous studies at the micro-level occurred in cases of an extremely high sampling density of 0.02–0.04 ha per sample for precision agriculture [11,12]. Therefore, there is a strong indication that soil sampling was overly detailed in these cases, whereas the accurate spatial representation of soil properties could be accomplished with sparser soil sampling and higher time- and cost-efficiency. The spatial autocorrelation had a secondary impact on interpolation accuracy, although a higher $R^2$ with the accuracy metrics for the $P_2O_5$ might indicate its suitability for the study of soil samples with lower spatial autocorrelation and higher variability. A similar approach to the application of Moran's I for the assessment of interpolation accuracy was implemented in the process of spatial prediction of the distribution of heavy metals [34]. This reinforces the assumption of its potential in cases of different soil properties, value ranges, and autocorrelation in comparison to this study, but also requires further research for specific conclusions.

Sampling density below 37.5% of the total soil sample set showed a considerable decline in the interpolation accuracy, indicating 13.7 ha per sample as the borderline sampling density in this study. This sampling density also corresponds to the breakpoint value of CV and autocorrelation, which both increased for sparser subsets. Most notably, spatial autocorrelation for 25% and 12.5% subsets showed an increase of up to 60% in value compared to denser subsets. This occurrence was similar for the sparsest sample in a study by Zhang et al. [10], which resulted in the lowest interpolation accuracy of the evaluated alternatives, as was the case in this study. OK and IDW achieved very close interpolation accuracy regarding soil sampling density. This observation is contrary to the majority of previous studies, which noted the superiority of geostatistical methods in

comparison to deterministic interpolation methods [1], or selected them a priori without evaluation [7]. However, this study indicates the importance of comparative assessment of interpolation accuracy using multiple complementary methods, such as OK and IDW. OK had the edge for interpolating higher autocorrelation and lower variability $K_2O$ values, while IDW achieved higher accuracy for the interpolation of $P_2O_5$ values and retained its local variability. This observation indicates the suitability of IDW for heterogeneous agricultural parcels with the presence of extreme values and lower spatial autocorrelation.

The influence of sampling density is expected to be similarly pronounced at the macro-level [16,18], compared to the micro-level observations represented by this study and multiple previous ones [12,14]. This statement is valid primarily for more homogeneous soils at the micro-level with a high level of spatial autocorrelation of a particular soil parameter [9]. In these cases, observing one or more agricultural parcels to determine precise fertilization could lead to less frequent sampling for the purpose of financial savings [35], which would potentially lead to a wider implementation of precise fertilization. For several agricultural parcels with similar tillage and crop rotation systems as the study area, it is recommended to perform an integrated soil sampling of the entire area. Such an approach reduces the financial burden on farmers and carries lower sampling density requirements for individual agricultural parcels. At the same time, this further expands the possibilities of geostatistical modeling, since the risk of an insufficient number of samples to join the mathematical model to the variogram with high reliability is reduced. For cases of sampling at the macro-level, in very few cases moderate or high spatial autocorrelation is achieved [17,18], making previous zoning and denser sampling in the locations of greater heterogeneity a recommended practice [7]. Previous research also indicates the importance of using an optimal sampling method concerning soil sampling quantity, with a regular grid being one of the most suitable soil sampling methods [17]. Furthermore, including auxiliary environmental variables could increase the interpolation accuracy of various soil parameters [36]. Future research is planned to implement independent predictors of soil parameters in order to implement high potential methods in spatial interpolation, such as regression kriging and machine learning [37].

### 5. Conclusions

Knowledge of the optimal soil sampling density increases the time- and cost-efficiency of the procedure, allowing the accurate spatial representation of soil parameters on both micro- and macro-levels. The performed analysis is expected to assist farmers in selecting optimal sampling density according to the desired level of detail while avoiding unnecessary costs. This approach could also prove useful for similar professions dependent upon time- and cost-intensive field sampling to reduce redundant fieldwork, such as biologists, geologists, and miners. Based on the research of the influence of spatial autocorrelation and sampling density on the interpolation accuracy using OK and IDW, it was determined that:

1. Interpolation accuracy primarily increases with the sampling density, having $R^2$ produced by linear regression in the range of 56.5–83.4%. Spatial autocorrelation indicated a lower impact on the interpolation accuracy but has potentially higher applicability in cases of lower spatial autocorrelation;
2. Both soil sampling density and spatial autocorrelation limit the interpolation accuracy if the number of input values is not large enough to accurately fit the mathematical model with a variogram for OK. In this study, sampling density below 37.5% on input data of 160 samples caused a rapid decrease in interpolation accuracy;
3. OK and IDW resulted in a similar interpolation accuracy for both soil $P_2O_5$ and $K_2O$ interpolation, while OK was more accurate in cases of lower CV and higher spatial autocorrelation. While deterministic interpolation methods, such as IDW, were inferior to OK in previous studies, they should be evaluated alongside geostatistical interpolation methods in similar studies.

Future research is planned to evaluate machine learning methods to predict soil parameters and the inclusion of environmental covariates. The rapid increase in the

popularity of these methods in recent years could have multiple benefits for farmers and land management planners, and it is necessary to optimize soil sampling density and methods according to the possibilities of the applied prediction methods.

**Author Contributions:** Conceptualization, D.R.; methodology, D.R.; software, D.R.; validation, I.J., V.V., M.J., M.G.; formal analysis, D.R.; investigation, D.R.; resources, D.R., I.J., V.V.; data curation, D.R.; writing—original draft preparation, D.R.; writing—review and editing, D.R., I.J., V.V., M.J., M.G.; visualization, D.R.; supervision, I.J., V.V., M.J., M.G.; project administration, M.J., M.G.; funding acquisition, D.R., M.J., M.G. All authors have read and agreed to the published version of the manuscript.

**Funding:** This research received no external funding.

**Acknowledgments:** This work was supported by the Faculty of Agrobiotechnical Sciences Osijek as a part of the scientific project: 'AgroGIT—technical and technological crop production systems, GIS and environment protection'. This work was supported by the University of Zagreb as a part of the scientific project: 'Advanced photogrammetry and remote sensing methods for environmental change monitoring' (Grant No. RS4ENVIRO).

**Conflicts of Interest:** The authors declare no conflict of interest.

## Appendix A

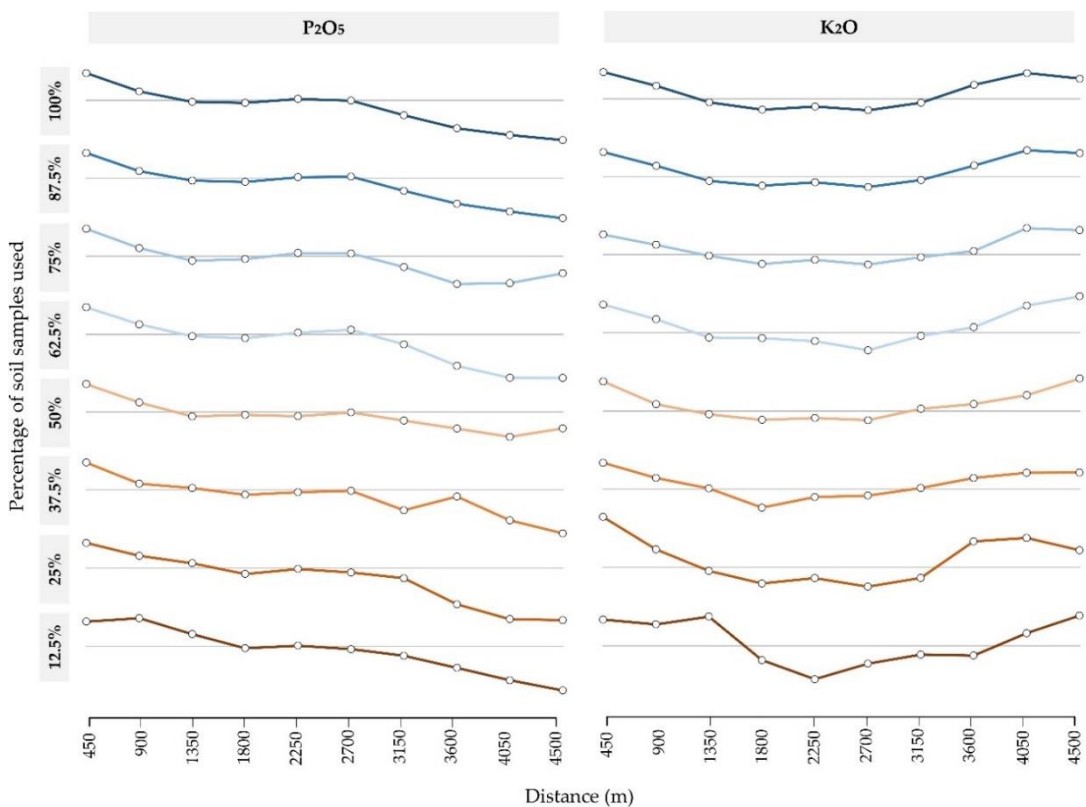

**Figure A1.** Correlograms of soil subsets, with the grey line indicating an autocorrelation value of zero.

**Table A1.** OK interpolation parameters for mathematical models with the highest coefficient of determination with variogram per input data set.

| Soil Property | Percentage of Soil Samples | n | s | r (m) | $R^2_v$ |
|---|---|---|---|---|---|
| P$_2$O$_5$ | 100% | 0.020 | 0.326 | 976 | 0.974 |
| | 87.5% | 0.024 | 0.326 | 1020 | 0.981 |
| | 75% | 0.055 | 0.536 | 937 | 0.964 |
| | 62.5% | 0.089 | 0.475 | 1151 | 0.869 |
| | 50% | 0.032 | 0.488 | 985 | 0.943 |
| | 37.5% | 0.012 | 0.349 | 1068 | 0.862 |
| | 25% | 0.011 | 0.358 | 1501 | 0.908 |
| | 12.5% | 0.016 | 0.104 | 1630 | 0.761 |
| K$_2$O | 100% | 0.159 | 0.397 | 1490 | 0.993 |
| | 87.5% | 0.017 | 0.242 | 1428 | 0.987 |
| | 75% | 0.020 | 0.238 | 1430 | 0.987 |
| | 62.5% | 0.006 | 0.235 | 1151 | 0.998 |
| | 50% | 0.058 | 0.461 | 985 | 0.951 |
| | 37.5% | 0.017 | 0.197 | 1068 | 0.747 |
| | 25% | 0.013 | 0.297 | 1651 | 0.745 |
| | 12.5% | 0.001 | 0.413 | 1585 | 0.759 |

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
