# Peer review of "The Effect of Soil Sampling Density and Spatial Autocorrelation on Interpolation Accuracy of Chemical Soil Properties in Arable Cropland"

_agronomy, doi:10.3390/agronomy11122430_

Round 1

Reviewer 1 Report

Dear Authors. The article is interesting, but very difficult to write. I was only able to read it because I needed to write a review. I repeat, the article is interesting and necessary, but your information is unlikely to be in demand, since the abundance of specific terms makes it impossible to study.

I earnestly ask you to add a few sentences, written without special terms, in the conclusion and in the summary. This will make your work interesting for everyone.

The first two sentences in the conclusion section are trivial, they are the most common words. If you take them away, the meaning of your work will not change.

Reviewer 2 Report

One point I would like clarified is the sample collection as in  normal soil sampling of agricultural  soils 10 to 20 soil cores are combined to from the sample within a field, is this the procedure used? 
